# Timing of Resource Addition Affects the Migration Behavior of Wood Decomposer Fungal Mycelia

**DOI:** 10.3390/jof7080654

**Published:** 2021-08-12

**Authors:** Yu Fukasawa, Koji Kaga

**Affiliations:** Kawatabi Field Science Center, Graduate School of Agricultural Science, Tohoku University, Miyagi 989-6711, Japan; kagamoku@gmail.com

**Keywords:** fungal behavior, directional memory, microbial intelligence, mycelial network, wood decay fungi

## Abstract

Studies of fungal behavior are essential for a better understanding of fungal-driven ecological processes. Here, we evaluated the effects of timing of resource (bait) addition on the behavior of fungal mycelia when it remains in the inoculum and when it migrates from it towards a bait, using cord-forming basidiomycetes. Experiments allowed mycelium to grow from an inoculum wood across the surface of a soil microcosm, where it encountered a new wood bait 14 or 98 d after the start of growth. After the 42-d colonization of the bait, inoculum and bait were individually moved to a dish containing fresh soil to determine whether the mycelia were able to grow out. When the inoculum and bait of mycelia baited after 14 d were transferred to new soil, there was 100% regrowth from both inoculum and bait in *Pholiota brunnescens* and *Phanerochaete velutina*, indicating that no migration occurred. However, when mycelium was baited after 98 d, 3 and 4 out of 10 replicates of *P. brunnescens* and *P. velutina*, respectively, regrew only from bait and not from inoculum, indicating migration. These results suggest that prolonged periods without new resources alter the behavior of mycelium, probably due to the exhaustion of resources.

## 1. Introduction

Fungi are fundamental agents in forest ecosystems due to their abilities to decompose organic matter and establish mycorrhizal and other types of mutualistic symbiosis with plants, as well as their pathogenicity to both plants and animals, which controls floral and faunal dynamics [1,2,3,4,5]. In particular, cord-forming basidiomycetes are important to relocate nutrients and carbon across their persistent linear organs––known as cords––which often form large networks at the interface of the litter layer and soil horizon in the forest floor [6,7,8]. Mycelial cord networks connecting numerous plant litter components––such as fallen trunks in case of saprotrophic species, and individual trees in case of mycorrhizal species––are abundant in the forest floor [9,10]. A better understanding of developmental cues, nutrient translocation, and mechanisms of network sustainability is essential to elucidate the dynamics of cycling and redistribution of carbon and other nutrients within the forest floor.

As each fungal species has its own abilities in terms of mycelial growth, organic matter decomposition, interaction with other organisms and so on, evaluating these traits for each single species is a promising line of inquiry for understanding fungal-driven ecosystem processes. In fungal ecology, a trait-based approach allows the accumulation of data and successfully improves ecosystem models, as seen in carbon sequestration models [11,12,13,14,15]. However, previous studies have focused on a relatively limited number of mycelial traits (e.g., extension rate, cord, density, and metabolite production [16]), and the behavioral characteristics of cord networks have been largely ignored. On the other hand, the behavior of cord-forming fungi has been well-studied for a limited number of species using soil tray microcosm experiments [9,17,18], introducing a new era of fungal ecology [19,20]. A better understanding of mycelial behavior is important for the advancement of trait-based fungal ecology.

One of the recent major findings in mycelial behavioral studies is that a mycelium can make a decision in response to environmental conditions [21,22]. When a wood block colonized by the cord-forming basidiomycete *Phanerochaete velutina* is placed as an inoculum on the surface of compressed unsterilized soil, the mycelium makes decisions on when, and with how much, intensity grows out from the inoculum onto the soil to search for new resources, based on information gathered from the remaining inoculum mass [22]. If a newly encountered resource (bait) is sufficiently large compared to the inoculum, the mycelium then makes decisions on whether to abandon the original inoculum in favor of the new one (i.e., through migration [21]). These results indicate that the mycelium of *P. velutina* makes behavioral decisions based on its economy across the entire mycelium and on the resources it occupies. This assumption would suggest that the quality of inoculum wood may also affect the decisions made by the mycelium. Furthermore, the mycelium of *P. velutina* retains memory of the direction towards which a new resource is located relative to the inoculum, when the inoculum is severed from the network and placed on fresh soil [21]. Polarized growth of fungal hyphae (cell-level directional memory) is well-studied [23]. Furthermore, other types of mycelial memory, such as temperature acclimation, have been reported in several fungal species [24,25,26,27,28,29,30]. However, mycelium-level directional memory has never been tested, other than in *P. velutina*.

In the present study, we aimed to test the effect of bait timing on mycelial decisions concerning migration to new wood resources and on memory of the direction of wood blocks to which it had been connected, if the cord connection between the inoculum and bait was completely destroyed. It was hypothesized that prolonged waiting would promote the mycelium’s decision to migrate to bait because of the depletion of resources in the inoculum and the demand for new resources. It was also hypothesized that directional memory would be promoted by waiting in the inoculum, because of the increased demand for new resources, but it would not affect the mycelium in the new bait because it may no longer need to maintain a link to the inoculum. The same set of experiments was applied to three species of cord-forming basidiomycetes: *Pholiota brunnescens*, *Phanerochaete velutina*, and *Resinicium bicolor*. The first species is a closely related ‘sister taxon’ of *Hypholoma fasciculare* [31,32], which is a cord-forming fungus regularly studied in previous research, as are the latter two species [9,17,18].

## 2. Methods

### 2.1. Fungal Culture and Wood Block Preparation

Kiln-dried beech (*Fagus crenata*) sapwood was cut into blocks of 1.5 cm × 1.5 cm × 1.5 cm (3.375 cm^3^) and dried at 70 °C to constant weight. Weighed blocks were numbered, soaked overnight in distilled H_2_O, and then autoclaved at 121 °C for 20 min in double-sealed autoclave bags. Autoclaving was repeated three times with 1-d intervals. The sterilized wood blocks were placed onto cultures of three different basidiomycetes: *Pholiota brunnescens* (NITE Biological Resource Center, NBRC culture collection, strain #110175), *Phanerochaete velutina* (#110184), and *Resinicium bicolor* (#110186). They were then grown on 0.5% malt extract agar (5 g L^−1^ malt extract, 15 g L^−1^ agar; Nakalai Tesque, Kyoto, Japan) in non-vented Petri dishes (2.5 cm thick, 14 cm in diameter). The dishes, including fungal strains and wood blocks, were sealed with Parafilm (Bemis Company Inc., Oshkosh, WI, USA) and incubated in the dark at 20 °C for 127 d before use in the soil microcosm experiment. In total, 60 inoculated blocks (20 blocks for each species) were prepared.

### 2.2. Microcosm Preparation

The soil was collected from the top 10 cm (A layer) in a deciduous mixed forest dominated by *Quercus serrata* and *Pinus densiflora* in Miyagi, Japan (38°37′ N, 140°48′ E, 129 m a.s.l.). After sieving it on site (through a 10 mm mesh), the soil was air-dried, sieved again through a 2 mm mesh, and frozen at –30 °C over 48 h to kill soil invertebrates. The soil was then rehydrated with DH_2_O (300 mL per 1 kg dried soil), transferred to Petri dishes (2.5 cm thick, 14 cm in diameter), and smoothed and compacted to about 5-mm thickness (approximately 60 g wet soil for each dish). One wood block, which was colonized by one of the three fungal species, had its surface cleaned by scraping mycelia and excess agar using a razor blade, and was placed at the center of each dish.

### 2.3. Microcosm Incubation

Each dish was weighed, and the lost water was replaced every week by spraying DH_2_O evenly across the soil surface until each dish reached its original mass. Dishes were covered with a lid, stacked, and sealed in polythene bags to reduce water loss, and were incubated at 20 °C in the dark. After 14 d (when mycelia had extended 3 cm from the inoculum in 50% of the dishes), a new beech wood block of the same size (bait; 1.5 cm × 1.5 cm × 1.5 cm)––prepared and sterilized, as described above, but not inoculated with fungi––was placed at the margin of the mycelium of the 10 of the 20 dishes for each fungal species (‘early-baited’ experiment). Another set of 10 dishes, which were not baited, was kept incubated and baited 84 d later (i.e., 98 d from start of the experiment) than the dishes of early-baited experiment dishes (‘wait’ experiment). Baited dishes were further incubated for 42 d, then both inoculum and bait wood blocks were retrieved, cleaned of surface mycelia, and placed centrally onto new soil dishes that were freshly prepared for each block, as described above. The dishes were further incubated at 20 °C for 7 d and the presence and location of outgrowing mycelium was recorded.

Dishes were randomly repositioned every 7 d during incubation to avoid possible effects of orientation and location within the incubator on the direction of hyphal growth. Dishes were photographed every 7 d, including each experimental occasion (baiting and transfer to a new soil tray), using a Canon EOS Kiss X5 camera, mounted on a stand at a height of 46 cm, and in the same light conditions to ensure consistency. After a 7-d incubation period on the new soil dish, both inoculum and bait blocks were harvested, cleaned of surface mycelia, dried at 70 °C to constant weight, and weighed. The weight losses of the inoculum wood blocks were calculated as follows:Weight loss (%) = (DW_t1_ − DW_t0_)/DW_t0_ × 100
where DW_t0_ is the dry weight of the block before it was incubated with the fungus, and DW_t1_ is the dry weight of the block after the experiment. As the DW_t0_ was not measured for each bait wood block, the weight loss of these was estimated by using the averaged DW_t0_ of the inoculum blocks.

### 2.4. Image Analysis

Photographs were analyzed using ImageJ (National Institute of Health, MD, USA). The length of one wood block side (1.5 cm) was used as a calibration ruler. The edge of each soil dish and the wood blocks were removed by windowing, and the resulting images were converted to black and white with a manually set threshold. The mycelia and soil were indicated by black and white pixels, respectively, allowing hyphal coverage (cm^2^) to be determined. Hyphal coverage was used as a measure of hyphal biomass on soil as it represents hyphal density in unit area [9,17,18]. To compare mycelial growth towards and away from the bait, each image was split into two at the central line of the wood block, based on methods by Fukasawa et al. [21].

### 2.5. Statistical Analysis

All statistical analyses were conducted using R 4.0.5 [33]. The time-series of total hyphal coverage and bait-side ratio of hyphal coverage were compared between the early-baited and wait experiments during a period from 7 d to 56 d by a repeated measures ANOVA, as the dish ID was set as a random effect (‘lmer’ function in *lme4* package and ‘anova’ function). A post-hoc comparison of the averages at each time point was conducted using the ‘lsmeans’ (in *lsmeans* package ver. 2.30-0), ‘pairs’, and ‘rbind’ functions with Sidak correction of the probability values. Significant differences from 0.5 in the bait-side ratio of hyphal coverage in the wait experiment were tested during the period from 98 d to 140 d using the Wilcoxon rank sum test with Bonferroni correction of probability values. Bait-side ratios of regrowing hyphal coverage on the new soil dishes were compared between early-baited and wait experiments for both inoculum and bait wood blocks using the Wilcoxon rank sum test.

The effects of four factors––inoculum and bait wood block weight losses (resource utilisation), coverage of hyphae regrowing from bait wood blocks (activity of mycelium in bait wood), and fungal species––on the presence/absence of regrowth from inoculum wood blocks on new soil dishes were evaluated using a generalized linear model (‘glm’ function). A binomial distribution error was assumed, and a logit link function was used. The best model was selected based on the Akaike information criterion by backward stepwise selection. The coefficients of explanatory variables in the best model were exponentiated to obtain odds ratios. Ratios > 1 indicated that the variables had a positive association with the presence of regrowth, while ratios < 1 indicated negative associations; the difference from 1 indicated the magnitude of the associations. The level of collinearity between predictor variables was evaluated by calculating the variance inflation factor (VIF); all VIF values were < 2, indicating low levels of multicollinearity in the model.

## 3. Results

The three fungal species displayed different morphologies while growing on soil (Figure 1, Appendix A): *P. brunnescens* formed fan-like structures, *P. velutina* grew in dense colonies, and *R. bicolor* formed thick but sparse cords that grew straight radially. The dynamics of soil surface coverage by the mycelium (hyphal coverage) were also unique to each fungal species (Figure 2). *P. brunnescens* showed a rhythmic growth as it stopped increasing hyphal coverage once between 21 and 28 d (Figure 2a). If a wood bait was not added on 14 d (wait experiment), the hyphal coverage of *P. brunnescens* started increasing again, and reached a maximum hyphal coverage of ~20 cm^2^ on day 49. The mycelium kept its hyphal coverage at a nearly maximum level until day 84, and then rapidly decreased. Hyphal coverage gradually decreased even after a wood bait was added on day 98. If a wood bait was added on day 14 (early-baited experiment), the increase in hyphal coverage was reduced, compared to that seen in wait mycelia, but the difference was not significant. The wait mycelium of *P. velutina* grew similarly to that of *P. brunnescens* and reached a maximum hyphal coverage of >20 cm^2^ on 35 d (Figure 2b). The mycelium kept its hyphal coverage at a nearly maximum level until day 70, with fluctuations due to its rhythmic ‘challenge and defeat’ behavior (Appendix A), and then decreased. The coverage also decreased even after a wood bait was added on day 98. However, if a wood bait was added on 14 d, the coverage of *P. velutina* was reduced significantly. In contrast, the mycelium of *R. bicolor* was less responsive to the addition of wood bait (Figure 2c). Its hyphal coverage in the wait experiment reached a maximum of ~7 cm^2^ on day 35 and then gradually decreased. It was only 1 cm^2^ when a wood bait was added to the wait mycelium. The bait-side ratio of hyphal coverage clearly increased after baiting in both the early-baited and wait mycelia of *P. velutina*, whereas *P. brunnescens* and *R. bicolor* were less responsive (Figure 3). The *P. velutina* mycelium present on the soil in the area of the inoculum wood block, but not connected to the bait, completely died back in the wait experiment (day 140, wait in Figure 1b), but showed lower mortality in the mycelium baited earlier (day 56, early-baited in Figure 1b).

After being transferred to new soil dishes, both inoculum and bait wood blocks of *P. brunnescens* and *P. velutina* in the early-baited system exhibited regrowth of mycelia, whereas none of the inoculum and bait wood blocks of *R. bicolor* showed regrowth (Appendix A; Figure 4). In the wait experiment, however, the presence/absence of regrowth from inoculum and bait wood blocks was different from that observed in the early-baited experiment (Fisher exact probability test, *p* < 0.001). In five and four replicates of *P. brunnescens* and *P. velutina*, respectively, mycelia never regrew from neither inoculum nor bait wood blocks. In three and four replicates of *P. brunnescens* and *P. velutina*, respectively, mycelia did not regrow from inocula but did regrow from bait wood blocks. Two replicates for both *P. brunnescens* and *P. velutina* mycelia regrew from both inoculum and bait wood blocks. In contrast, the mycelia of *R. bicolor* never regrew from them in neither early-baited nor wait experiments. The hyphal coverage regrown from inoculum wood blocks was larger in early-baited than in wait experiments for *P. brunnescens* and *P. velutina* (Figure 5). The hyphal coverage regrown from bait wood blocks was also significantly larger in the early-baited system of *P. velutina*, but not in that of *P. brunnescens*. The hyphal coverage ratio of the bait-side of the *P. brunnescens* inoculum was significantly larger than 0.5 (Figure 6). In contrast, that of the *P. velutina* inoculum and bait wood blocks of *P. brunnescens* and *P. velutina* were not significantly different from 0.5. The weight losses of inoculum wood blocks were larger in the wait system than in the early-baited system in all three fungal species tested, whereas those of bait wood blocks were not significantly different between the two systems (Figure 7). Among the four variables tested, fungal species and weight loss of inoculum wood blocks were selected in the best GLM model (AIC = 30.885) to explain regrowth from inoculum on new soil dishes. Weight loss of inoculum wood block was negatively associated (odds ratio = 0.795) with the presence of regrowth from the inoculum (Figure 8).

## 4. Discussion

### 4.1. Effects of Bait Addition Timing on the Mycelial Decision to Migrate

As predicted, the results showed that, when the mycelia of *P. brunnescens* and *P. velutina* grew from inoculum wood blocks and colonized new bait of the same size, if the inoculum wood was already substantially decayed (up to 50% and 60% in *P. brunnescens* and *P. velutina*, respectively; Figure 8), the mycelia were often no longer able to grow out of the original inoculum, and grew out from the newly colonized bait wood block instead. This behavior represents the migration of mycelia from original inoculum to bait wood block [21]. However, such migration was not observed if the inoculum wood was not yet substantially decayed (up to 30% in both *P. brunnescens* and *P. velutina*), and regrowth from both inoculum and bait wood were recorded (Figure 8). These results suggest that the mycelia of *P. brunnescens* and *P. velutina* may recognize the percentage of the resources remaining in the original inoculum and change their behavior to abandon the inoculum wood block in favor of new bait. This is in line with our previous observations of the mycelium of *P. velutina*, which showed that the intensity of hyphal outgrowth onto the soil from inoculum wood blocks depended on the percentage weight loss of the inoculum [22], and that the occurrence of migration depended on the amount of the new resources the mycelia encountered [21].

Interestingly, in the wait mycelia of *P. velutina*, hyphal coverage showed a rhythmic expansion and reduction after it reached maximum on day 35 until it began to decrease on day 70 (Figure 2, Appendix A). This is perhaps the recycling of mycelium and renewed growth to re-start searching. Although recycling probably involves autophagy and apoptotic-like mechanisms, it is not clear how changes are made in order to trigger autophagy and renewed growth [34]. Previous studies have demonstrated bidirectional translocation of materials across mycelial networks, both acropetal and basipetal, based on local demand within the mycelium [35,36,37,38,39,40,41,42]. However, the underlying mechanisms of the rhythmic expansion–reduction–re-expansion cycle observed in the present study may be different from those of relatively short-term (10–60 h) oscillatory material transfers observed in the previous studies [40,43,44], because the former cycle is considerably longer (2–4 weeks). When does a mycelium switch its protoplasmic flow from forward to reverse and vice versa? An important insight on this issue was obtained from the experiments of Tlalka et al. [43], which reported that the nitrogen transfer was strongly polarized in a developing juvenile colony of *P. velutina* (10–13 d old), but that once it matured (5-week old), the ‘phase’ changed and bidirectional transport started. The mechanisms determining the timing of transport phase changes are not known [34,43]. As discussed in the above section, a possible factor determining such long-term growth dynamics is the amount of resources a mycelium holds within its hyphal body or in the wood material that it occupies [22]. Currently, however, there is insufficient fungal evidence to answer this question. Given that slime mold plasmodia have a superficially similar body design to that of fungal mycelia and that the foraging behavior of plasmodia looks quite similar to that observed in the present study, the published literature on the behavior of slime molds might shed some light on this question. Dussutour et al. [45] found that the carbon to nitrogen ratio is a key factor determining growth decisions in plasmodium. A plasmodium can alter its growth form and movement to regulate the supply of carbon and nitrogen to a target ratio that maximises performance, and can make ‘decisions’ to select multiple baits with different qualities, if there is a choice. It will be interesting to test the effect of wood inoculum quality (C/N ratio), available nutrient amounts in soil, and the addition of a labile carbon source (e.g., glucose) on the variation of mycelial growth phases and migration.

The present study indicated the possibility that resource quality affects mycelial migration. However, another factor potentially affecting mycelial migration is time duration. The wait mycelia spent considerably more time (seven-fold longer) than early-baited ones before baiting (Figure 1). Given that unsterilized soil was used in the microcosm, remaining for a longer period of time on soil meant a longer competition against soil microorganisms. We observed that soil fungi, such as *Chaetomium globosum* and *Trichoderma* spp., colonized and sporulated on some wood blocks during the experiment, thus many other invisible microscopic soil fungi might also be competitors for the focal fungal strains. In this case, the antagonistic response of fungal mycelia includes the production of various secondary metabolites [46,47,48], which might be energetically expensive for focal fungal mycelia; such a physiological challenge is quite likely to alter mycelial behavior [49]. Although the production of secondary metabolites was not measured in the present study, yellow/brown pigmentation was observed on the mycelial cords of *P. brunnescens* and *P. velutina* (Figure 1). Such a balance between costs and benefits to protect the wood from surrounding competitors could be a primary factor affecting the mycelial behavior to fully migrate to new resources, as Fukasawa et al. [21] reported that the mycelium of *P. velutina* left the smallest inoculum more often than it left larger inocula, if their wood quality was uniform. Again, a slime mold study indicated a possible explanation of this behavior. Takamatsu et al. [50] reported that a plasmodium formed a more ‘exploring’ morphology (sparsely spreading dendrites with thick tubes) in the presence of repellent in the system, while it formed a ‘stay’ morphology (round shape characterized by a thin sheet structure with unclear thin tubular structures) in favorable conditions. The absence of regrowth from both inoculum and bait wood blocks in five and four wait experiments on *P. brunnescens* and *P. velutina*, respectively (Figure 4), indicated that the timing of baiting employed in the present study was barely sufficient to allow the mycelia of these two species to remain active, as their hyphal coverages were already beginning to decrease (Figure 2). To remove such a potential effect of time duration on competition and associated antagonistic reactions, further studies using inoculum wood blocks with various qualities with equal incubation periods are needed.

In contrast to *P. brunnescens* and *P. velutina*, the mycelia of *R. bicolor* did not show regrowth from any of the inoculum and bait wood blocks, even though the weight losses of those blocks were less than 40%. A possible explanation for this is the difference in growth durations, competitive abilities, and distinct nutritional requirements among the three fungal strains. DNA metabarcoding studies found that *R. bicolor* dominates early stages of deadwood decomposition [51]. Competition studies reported that *R. bicolor* is less competitive than *P. velutina* and *Hypholoma fasciculare* (a closely related ‘sister taxon’ of the genus *Pholiota* used in the present study [31,32]) in stressful conditions, such as high temperature, and under invertebrate grazing pressure [47,52,53,54,55,56]. These results suggest that *R. bicolor* is a weak competitor in stressful field conditions, and is a ruderal species that starts colonization immediately after a tree dies, utilizing only a small fraction of wood, and moving to new resources quickly. Our upcoming data from pure culture study show that these three fungal strains caused mass losses on wood blocks not significantly different each other. However, even if the mass loss of wood blocks were not different among the strains, the amount of nutrients available for them could be different, depending on their potentially distinct nutritional requirements. No attempts were made to reisolate the fungus from the original inoculum and bait, so it is not certain whether the fungus had completely lost its viability within the wood blocks. However, the observations certainly suggest that mycelia were more or less inactive when they did not regrow from the wood––as the wood color darkened without any visible hyphae on the surface––and it was likely that the focal fungus was replaced by soil fungi.

### 4.2. Mycelial Memory of the Direction of Growth

On the new soil dish, *P. brunnescens* mycelia showed a dominant regrowth from the inoculum side that had originally been linked to the bait on the original soil dish. This is known as directional growth memory, and was previously reported in *P. velutina* [21]. A difference observed between *P. brunnescens* and *P. velutina* was that the former did not apparently show any visible reallocation of mycelial biomass and mycelial growth in the direction of the bait on the original soil dish, like the latter did (Figure 1 and Figure 3, [21]). This indicates that the dominance of mycelia in that direction may not be necessary to allow directional memory to be retained on the new soil dish. However, it is possible that cytoplasm was transported towards the bait direction leaving empty hypha, which are visible in unconnected areas in the obtained photographs. Although *P. velutina* mycelia showed directional memory in a previous study [21], this species did not regrow to a greater extent from the side that originally faced to the new bait in the present study. This is not surprising because, even in Fukasawa et al. [21], the mycelia of *P. velutina* did not necessarily show directional memory in all combinations of inoculum and bait. Previous hyphal-level studies suggested that intensity of directional memory varied depending on fungal species [57,58,59]. The conditions in which a fungal mycelium shows directional growth memory may also vary among fungal species.

In spite of the recent advances in the memory storage mechanisms of polarized hyphal tip growth, which are regulated by the location of Spitzenkörper (an assembly of vesicles at the hyphal tip) and cytoskeleton [23], the way that information related to environmental conditions is stored and reflected in mycelial behavior is largely unknown [19,20]. A possible explanation of directional memory observed in the present study is simply the development of more mycelium in one part of the wood block than in the opposite side. Another important aspect was reported in a study by Tlalka et al. [43], where a map of locally synchronised oscillatory phase domains of nitrogen transfer across a mycelium of *P. velutina* was created. The results showed that the hyphae connecting the original inoculum and bait wood blocks presented a slightly different oscillation phase compared to the rest of the mycelium. If such a gradient in hyphal physiological activity was kept within a mycelium, it will be maintained during the regrowth in the new soil tray. As only a couple of wait inocula showed regrowth of hyphae on new soil dishes, it was not possible to statistically test whether the directional memory was kept after waiting for bait (Figure 6). However, two replicates in the wait mycelia of *P. velutina* showed large values close to 1. This is likely due to the mycelium that remained in the side of the inoculum wood close to the bait before it departed towards the bait.

To summarize, our results showed a possibility that timing of resource addition affects mycelial migration behavior. We appreciate that there may be semantic conflicts in the concept of fungal behavior among scientists as it is a novel and developing study field. However, we believe that recognizing such a growth response of fungal mycelia to internal and external resource supply as a behavior is a first step in the study of mycelial behavior. The results raised new questions regarding which of available resource amount and time duration affect mycelial behavior, and which should be tested in the future. For example, to evaluate the effect of bait on mycelial migration more accurately in the wait experiment, control dishes without bait are also needed. The presence/absence of bait after the long waiting period without bait influences the regrowth from inoculum wood after the transfer may indicate a decision of mycelium to migrate. However, if presence/absence of bait does not affect the regrowth, there will be no decision by mycelium and just a die back of mycelium following complete depletion of the available resources. Concerning the direction of the regrowth of the fungi, a trend of growth from the inoculum side linked to bait was noticed for one fungal strain only, which does not allow any strong general conclusions to be drawn on memory of fungal mycelia.

## Figures and Tables

**Figure 1 jof-07-00654-f001:**
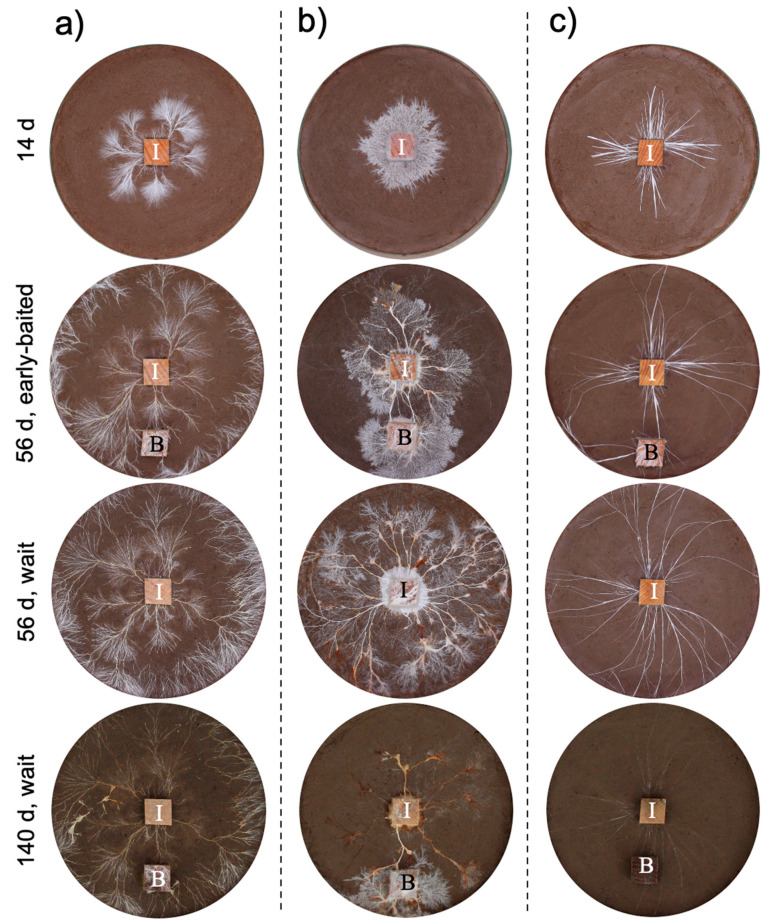
The effect of added resources on the development of (**a**) *Pholiota brunnescens*, (**b**) *Phanerochaete velutina*, and (**c**) *Resinicium bicolor*. Representative images of mycelial systems grown from beech wood blocks (I) on non-sterile compressed soil in Petri dishes (576 cm^2^) are shown at the time points indicated for ‘early-baited’ and ‘wait’ treatments. B indicates bait wood blocks. One side length of each wood block is 1.5 cm.

**Figure 2 jof-07-00654-f002:**
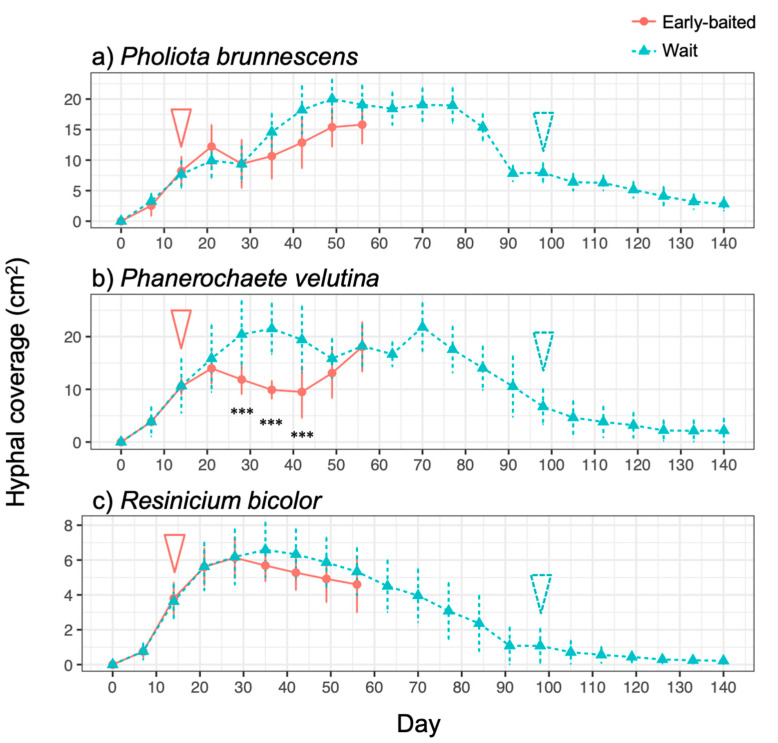
Change in hyphal coverage of (**a**) *Pholiota brunnescens*, (**b**) *Phanerochaete velutina*, and (**c**) *Resinicium bicolor* in response to added resources at different times. Pink round plots with solid line indicate the ‘early-baited’ experiment, while blue triangle plots with a dotted line indicate the ‘wait’ experiment. Pink solid arrow and blue dotted arrow indicate the time of addition of bait wood blocks to ‘early-baited’ and ‘wait’ experiments, respectively. Plots and error bars indicate mean and standard deviation, respectively (*N* = 10). Asterisks indicate significant differences between ‘early-baited’ and ‘wait’ experiments: *** *p* < 0.001 (repeated measures ANOVA).

**Figure 3 jof-07-00654-f003:**
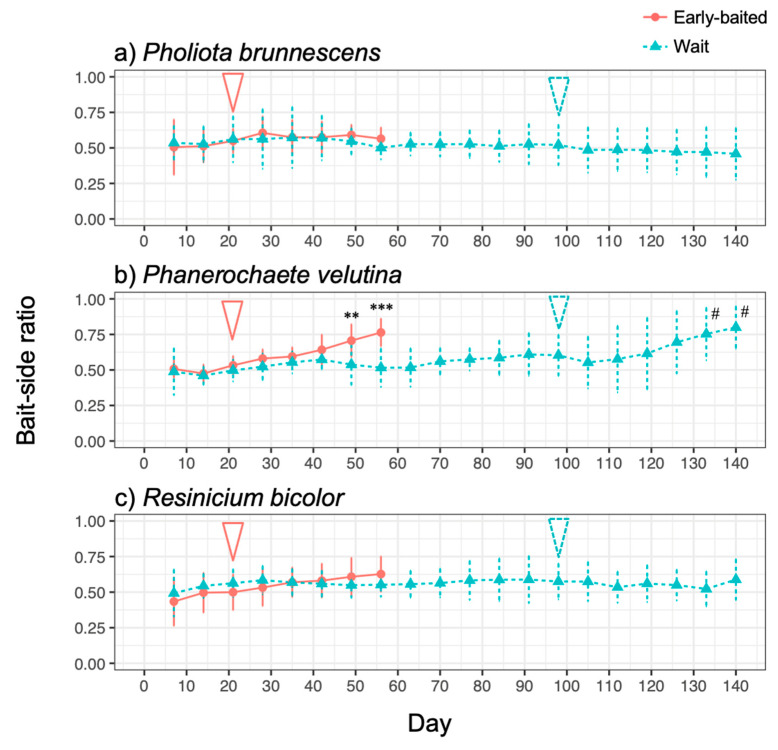
Changes in the bait-side ratio of the hyphal coverage of (**a**) *Pholiota brunnescens*, (**b**) *Phanerochaete velutina*, and (**c**) *Resinicium bicolor* in response to added resources at different times. Pink round plots with a solid line indicate the ‘early-baited’ experiment and blue triangle plots with dotted line indicate the ‘wait’ experiment. The pink solid arrow and blue dotted arrow indicate the time of addition of bait wood blocks to ‘early-baited’ and ‘wait’ experiments, respectively. Plots and error bars indicate mean and standard deviation, respectively (*N* = 10). Asterisks indicate significant differences between ‘early-baited’ and ‘wait’ experiments during 7–56 d: ** *p* < 0.01; *** *p* < 0.001 (repeated measures ANOVA). Hashtags indicate significant difference from 0.5 in wait experiment (Wilcoxon rank-sum test: # *p* < 0.05).

**Figure 4 jof-07-00654-f004:**
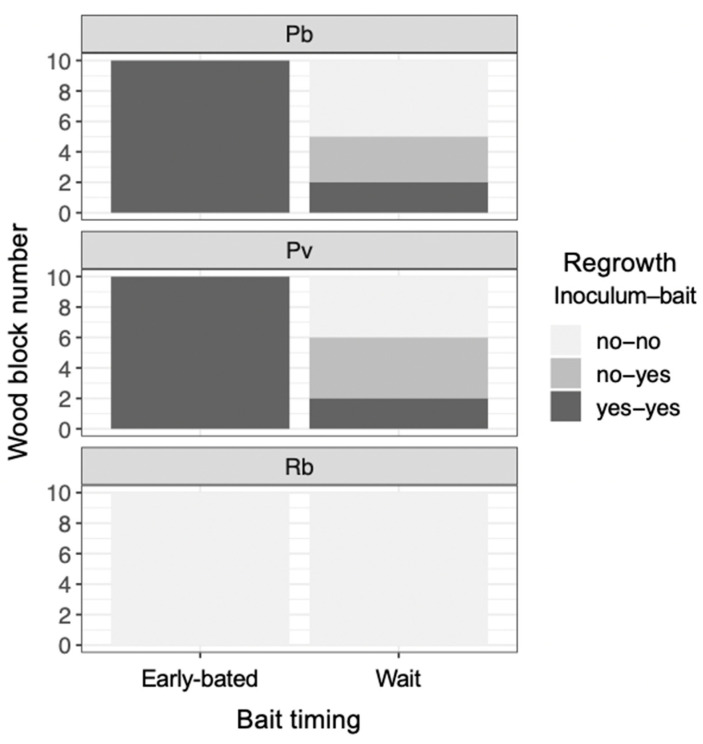
The frequency of wood blocks (inoculum and bait) with/without hyphal regrowth 7 d after the blocks had been transferred to new soil dishes, depending on the timing of bait added. Abbreviations: Pb, *Pholiota brunnescens*; Pv, *Phanerochaete velutina*; Rb, *Resinicium bicolor*. Note that the frequency of regrowth from inoculum and bait wood blocks were different between ‘early-baited’ and ‘wait’ experiments (Fisher exact probability test, *p* < 0.001).

**Figure 5 jof-07-00654-f005:**
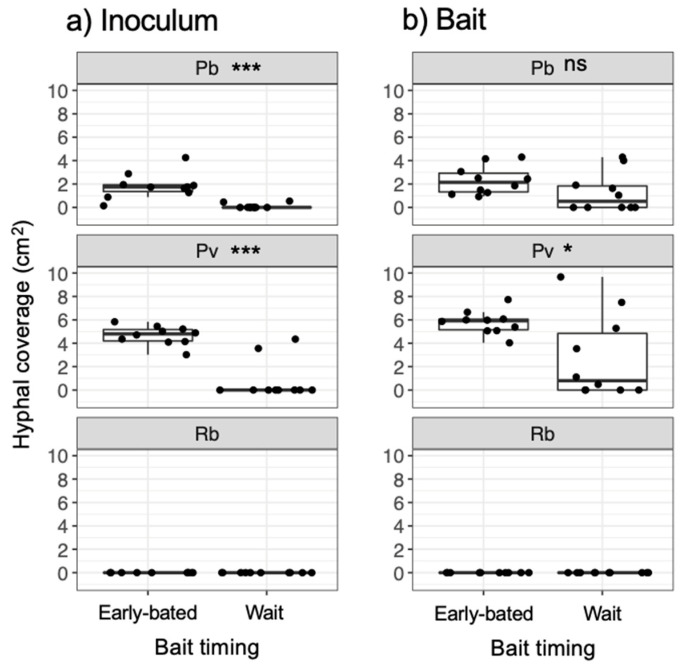
Hyphal coverage (cm^2^) of mycelia extending from (**a**) inoculum and (**b**) bait wood blocks 7 d after they had been transferred to new soil dishes, depending on the timing of bait added. Abbreviations: Pb, *Pholiota brunnescens*; Pv, *Phanerochaete velutina*; Rb, *Resinicium bicolor*. Asterisks indicate significant differences between ‘early-baited’ and ‘wait’ experiments (Wilcoxon rank-sum test: * *p* < 0.05; *** *p* < 0.001; ns, not significant, *N* = 10). Balck dots are individual data. Upper and lower ends of vertical lines indicate maximum and minimum values, respectively. The tick horizontal line indicates median.

**Figure 6 jof-07-00654-f006:**
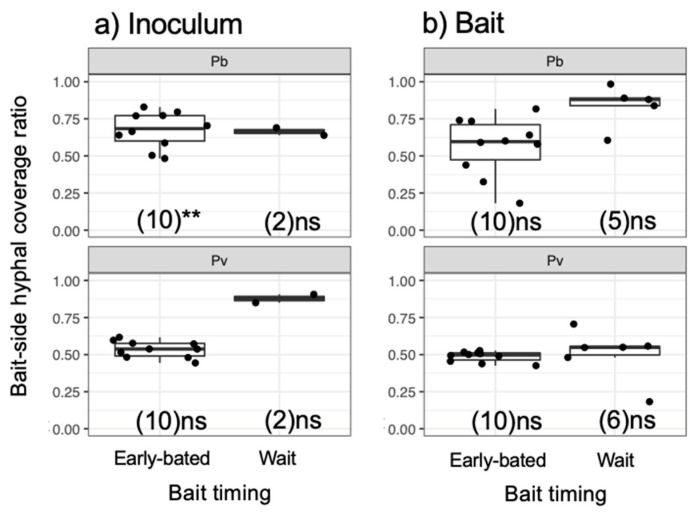
Bait-side hyphal coverage ratio against the whole mycelium coverage (bait-side + opposite side) of (**a**) inoculum and (**b**) bait wood blocks. Abbreviations: Pb, *Pholiota brunnescens*; Pv, *Phanerochaete velutina*. Asterisks indicate significant differences from 0.5 (Wilcoxon rank-sum test: ** *p* < 0.01; ns, not significant). Numbers of replicates were shown in parenthesis. Balck dots are individual data. Upper and lower ends of vertical lines indicate maximum and minimum values, respectively. The tick horizontal line indicates median.

**Figure 7 jof-07-00654-f007:**
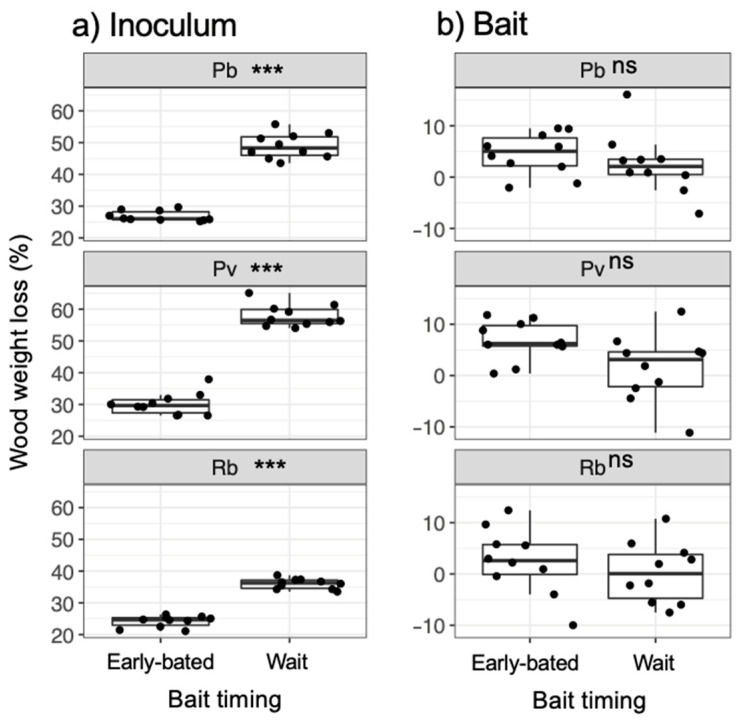
Weight loss of (**a**) inoculum and (**b**) bait wood blocks. Abbreviations: Pb, *Pholiota brunnescens*; Pv, *Phanerochaete velutina*; Rb, *Resinicium bicolor*. Asterisks indicate significant differences between ‘early-baited’ and ‘wait’ experiments (Wilcoxon rank-sum test: *** *p* < 0.001; ns, not significant, *N* = 10). Balck dots are individual data. Upper and lower ends of vertical lines indicate maximum and minimum values, respectively. The tick horizontal line indicates median.

**Figure 8 jof-07-00654-f008:**
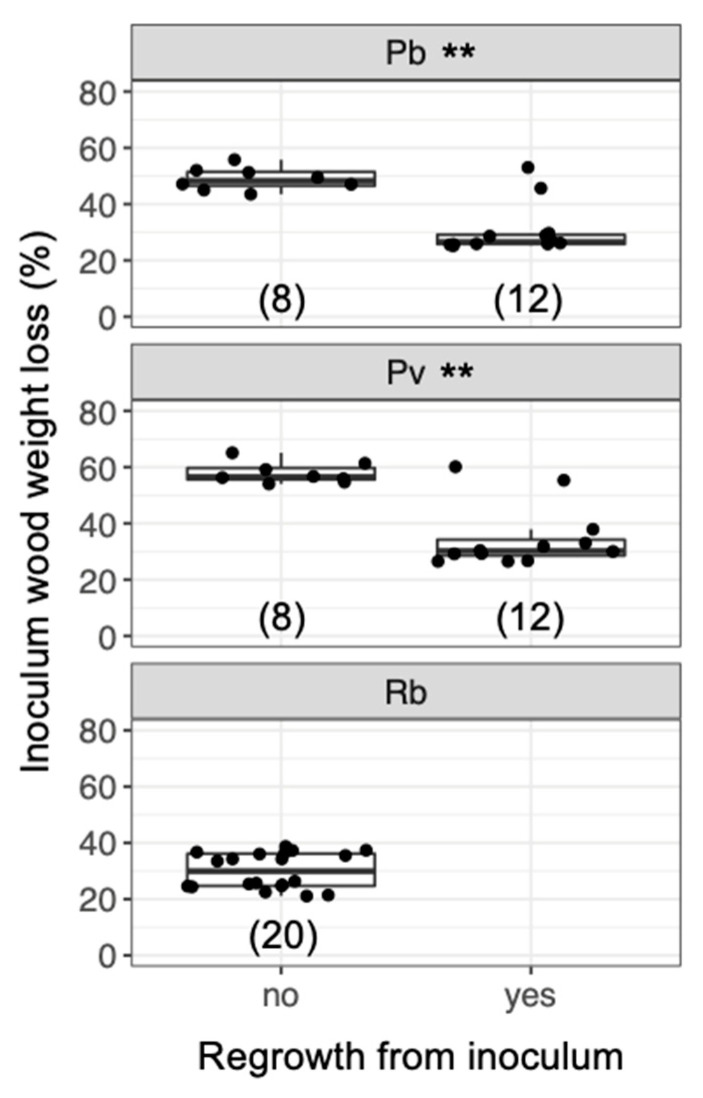
Weight loss of inoculum wood blocks with/without hyphal regrowth 7 d after the blocks had been transferred to new soil dishes, regardless of the timing of added bait. Abbreviations: Pb, *Pholiota brunnescens*; Pv, *Phanerochaete velutina*; Rb, *Resinicium bicolor*. Asterisks indicate significant differences between wood blocks with and without hyphal regrowth (Wilcoxon rank-sum test: ** *p* < 0.01). Numbers of replicates are shown in parenthesis. Balck dots are individual data. Upper and lower ends of vertical lines indicate maximum and minimum values, respectively. The tick horizontal line indicates median.

## Data Availability

The data presented in this study are available on request from the corresponding author.

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
