# Peer review of "Timing of Resource Addition Affects the Migration Behavior of Wood Decomposer Fungal Mycelia"

_jof, 2021, doi:10.3390/jof7080654_

Round 1
Reviewer 1 Report
I reviewed the manuscript entitled “Waiting affects the behavioural decision of wood decomposer fungal mycelia” submitted by Yu Fukasawa and Koji Kaga to Journal of Fungi.
The authors raised an interesting and largely under-studied question in fungal ecology, which is the behavioural characteristics of fungi. Even if the results do not allow a clear answer to all the questions asked by the study, I think that this paper constitutes a highly interesting premise, which should generate further studies on this under-considered subject. This study deals with a new area of prospection that could bring valuable findings about fungal ecology based on fungal traits, and so I think that the paper would fit very well into the scope of Journal of Fungi.
A sterilized wood block inoculated with a fungal strain, referred to as “inoculum”, is grown on a soil microcosm. Then, after 14 or 98 days, a new sterilized wood block, referred to as “bait”, is added in the same microcosm for a period of 42 days. At the end of this period, both wood blocks are moved individually in a new, fresh, soil microcosm for 7 days. The authors performed the experiment with three different fungal strains.
The ability of the inoculated fungi to grow/regrow on/from inoculum and bait (i.e., ability to migrate from the inoculum to the bait) and the direction in which the fungal mycelium grows (i.e., ability to memorize the direction of the bait) are studied.
The authors found that, when the bait is brought after 14 days, the fungi is able to regrow on the new soil microcosm from both the inoculated wood and the bait, while when the bait is brought after 98 days, the regrowth is observed mainly from the bait for two of the three fungal strains. Concerning the direction of the regrowth of the fungi, a trend of growth from the inoculum side linked to the bait is noticed for one fungal strain only, which does not allow conclusions to be drawn.
I would suggest a few revisions to clarify some points:
- Title: I would replace the term “waiting” by a more precise term to make more comprehensive the title and to be more appealing for the readers. For example, “Timing of resource addition” or something like that.
Methods:
- l.41: correct “improvs”
- l.82: Could you please precise which part of the wood you used (branch, trunk, … and heartwood, sapwood,…)?
- l.89 to 93: I do not understand if you are describing here the conditions in which the fungal strains have been cultivated prior to their use for inoculation of the wood blocks, or if you are describing the conditions for inoculation of the wood blocks themselves. Could you please be clearer in the text?
If you speak about the culture of the fungal strains prior to their use for inoculation of the wood blocks, please consider adding in the Methods the details on the conditions and duration necessary to inoculate the wood blocks.
- l.92: If you are describing here the inoculation of the wood blocks, I was wondering why you waited such a long time (i.e., 127 days) for the inoculation? Were the wood blocks not already partially decomposed? (It does not seem from the pictures but this is surprising, because the wood blocks are quite small). Were there significant differences in initial mass of inoculum wood blocks (i.e., after initial inoculation) between the three fungal strains?
In addition, if you used this same duration of culture (127 days) for inoculation of the wood with the three different fungal strains, does it can have an effect on the further need to find a new resource in the soil microcosm experiment? I mean, the three fungal strains may have distinct growth durations and distinct nutritional requirements. Maybe R. bicolor used almost all the nutrients that it was able to metabolize from the wood block during the 127 days, and so when the experiment is beginning it has not enough energy to efficiently colonize the bait. On the contrary, the other fungal species that can use more diversified wood components may have more energy at the beginning of the experiment to overcome competition and colonize bait.
- l.97: Did you study the soil chemistry? Do you think that it can differentially affect the passage from the inoculum to the bait according to the potentially different nutritional needs of the fungal strains?
- l.100: You say in the discussion that the soil microbes can affect the growth of the fungi due to competition (see lines 317-318), so why did you not sterilize the soil to remove the microbes as well? Did you observe the colonization of some wood blocks with another fungal strain coming from the soil?
- l.115: Do you mean “98 d” instead of “94 d”?
- l.160: Could you please precise here the R function used for the generalized linear models, as you did in the first paragraph for the repeated measures analyses?
Results:
-l.194-195: Could you please add on the Figure 1, into the different panels, the letters you are mentioning in the text?
- l.218: Is it possible that the size of the Petri dish was limiting for R. bicolor growth? Because it seems from the supplementary pictures that this fungus had a linear growth, linear morphology, and it was the first to colonize the entire Petri dish.
- l.357: It would be interesting that the authors show some pictures of soil microcosms at the end of the experiment, i.e., the new soil dishes after the 7 days incubation, because many results derived from this last part of the experiment.
Author Response
The authors raised an interesting and largely under-studied question in fungal ecology, which is the behavioural characteristics of fungi. Even if the results do not allow a clear answer to all the questions asked by the study, I think that this paper constitutes a highly interesting premise, which should generate further studies on this under-considered subject. This study deals with a new area of prospection that could bring valuable findings about fungal ecology based on fungal traits, and so I think that the paper would fit very well into the scope of Journal of Fungi.
Thank you for your extremely positive comments.
The authors found that, when the bait is brought after 14 days, the fungi is able to regrow on the new soil microcosm from both the inoculated wood and the bait, while when the bait is brought after 98 days, the regrowth is observed mainly from the bait for two of the three fungal strains. Concerning the direction of the regrowth of the fungi, a trend of growth from the inoculum side linked to the bait is noticed for one fungal strain only, which does not allow conclusions to be drawn.
We added a short concluding paragraph at the end of the Discussion:
“To summarise, our results showed a possibility that timing of resource addition affects mycelial migration behaviour. We appreciate that there may be semantic conflicts in the concept of fungal behaviour among scientists as it is a novel and developing study field. But we believe that recognizing such a growth response of fungal mycelia to internal and external resource supply as a behaviour is a first step in the study of mycelial behaviour. The results raised a new question that which of available resource amount and time duration affect mycelial behaviour, which should be tested in the future. Concerning the direction of the regrowth of the fungi, a trend of growth from the inoculum side linked to the bait was noticed for one fungal strain only, which does not allow any strong general conclusions to be drawn on memory of fungal mycelia.”
- Title: I would replace the term “waiting” by a more precise term to make more comprehensive the title and to be more appealing for the readers. For example, “Timing of resource addition” or something like that.
We replaced the term “waiting” by “timing of resource addition”. Also we change the title to calm down our arguments for fungal decision.
Methods:
- l.41: correct “improvs”
Corrected.
- l.82: Could you please precise which part of the wood you used (branch, trunk, … and heartwood, sapwood,…)?
We added “sapwood” here.
- l.89 to 93: I do not understand if you are describing here the conditions in which the fungal strains have been cultivated prior to their use for inoculation of the wood blocks, or if you are describing the conditions for inoculation of the wood blocks themselves. Could you please be clearer in the text?
If you speak about the culture of the fungal strains prior to their use for inoculation of the wood blocks, please consider adding in the Methods the details on the conditions and duration necessary to inoculate the wood blocks.
Yes, these descriptions are for fungal inoculation on wood blocks. We deleted a sentence about the origin of fungal strains to clarify the connection of the sentences.
- l.92: If you are describing here the inoculation of the wood blocks, I was wondering why you waited such a long time (i.e., 127 days) for the inoculation? Were the wood blocks not already partially decomposed? (It does not seem from the pictures but this is surprising, because the wood blocks are quite small). Were there significant differences in initial mass of inoculum wood blocks (i.e., after initial inoculation) between the three fungal strains?
In our previous paper (Fukasawa and Kaga 2020. Effects of wood resource size and decomposition on hyphal outgrowth of a cord-forming basidiomycete, Phanerochaete velutina. Scientific Reports 10: 21936), we observed less hyphal growth on soil and wood mass loss than we expected after 2 months of colonization period. Since hyphal outgrowth was associated with wood mass loss, we made the inoculation period double (4 months) to achieve sufficient hyphal outgrowth in the present study. Our upcoming data from pure culture study showed that these three fungal strains caused only a small weight loss (less than 5%) on 2x2x2 cm beech wood blocks after 3 months incubation on 0.5% Malt extract agar medium and the weight losses were not significantly different among the strains.
In addition, if you used this same duration of culture (127 days) for inoculation of the wood with the three different fungal strains, does it can have an effect on the further need to find a new resource in the soil microcosm experiment? I mean, the three fungal strains may have distinct growth durations and distinct nutritional requirements. Maybe R. bicolor used almost all the nutrients that it was able to metabolize from the wood block during the 127 days, and so when the experiment is beginning it has not enough energy to efficiently colonize the bait. On the contrary, the other fungal species that can use more diversified wood components may have more energy at the beginning of the experiment to overcome competition and colonize bait.
Yes, your point is important for explaining difference in the effect of baiting timing on mycelial behaviour among fungal species. Even if the mass loss of wood blocks were not different among the three fungal species, amount of nutrients available for different fungal species may differ depending on their nutritional requirements. We added related discussion at the end of the first section (4.1) of Discussion.
- l.97: Did you study the soil chemistry? Do you think that it can differentially affect the passage from the inoculum to the bait according to the potentially different nutritional needs of the fungal strains?
No, we did not measure soil chemistry, and yes, we think that soil nutrition affects mycelial passage from the inoculum to the bait according to the potentially different nutritional needs of the fungal strains. We added a related statement about further study in the end of the second paragraph in Discussion.
- l.100: You say in the discussion that the soil microbes can affect the growth of the fungi due to competition (see lines 317-318), so why did you not sterilize the soil to remove the microbes as well? Did you observe the colonization of some wood blocks with another fungal strain coming from the soil?
We did not sterilize the soil in the microcosm because the presence of soil microorganisms as a competitor for the focal strains is important for mycelial migration from inoculum to bait. If soil was sterilized and only a focal strain was living in the microcosm, mycelium do not need to leave original inoculum even after it lost its nutritional value to defend. We observed Chaetomium globosum and Trichoderma spp. colonizing on some of the wood blocks. We added a related discussion in the third paragraph in 4.1.
- l.115: Do you mean “98 d” instead of “94 d”?
Yes, you are right. The date was corrected.
- l.160: Could you please precise here the R function used for the generalized linear models, as you did in the first paragraph for the repeated measures analyses?
We added “(‘glm’ function)” here.
Results:
-l.194-195: Could you please add on the Figure 1, into the different panels, the letters you are mentioning in the text?
We corrected the figure citation in the text and added a vertical line in Figure 1 to separate images clearly.
- l.218: Is it possible that the size of the Petri dish was limiting for R. bicolor growth? Because it seems from the supplementary pictures that this fungus had a linear growth, linear morphology, and it was the first to colonize the entire Petri dish.
Yes, the size of the Petri dish could limit the maximum hyphal cover of R. bicolor, but it may not affect the regrowth from wood blocks.
- l.357: It would be interesting that the authors show some pictures of soil microcosms at the end of the experiment, i.e., the new soil dishes after the 7 days incubation, because many results derived from this last part of the experiment.
We added a new Figure S2 in a supplementary file to show pictures of soil microcosms on the new soil dishes.

Reviewer 2 Report
Review jof-1320447
General comments
The manuscript presents novel data on three cord-forming Basidiomycetes, that are grown from wood inocula and exposed to “baits” (=fresh wood blocks) after differing time periods, 14 or 98 days. The authors argue that after a longer period the fungi “decide” to migrate to the new bait and leave the inoculum (which is shown by regrowth from the wood blocks), whereas after shorter periods regrowth occurs from both, inoculum and bait wood blocks.
The manuscript is very well written, and the results analyzed and presented nicely. This study follows several recent studies, partly by the authors, on the topic of “fungal behaviour”, that analysed the growth of cord-forming fungi in relation to inocula and baits, and the “decision making” mostly based on differential sizes of baits and inocula. This study clearly differentiates in its topic based on the novel idea to introduce timing as a factor, and using three fungal species instead of only one.
These previous studies indeed proved some interesting patterns, especially that fungi cannot be isolated from an inoculum wood in case the new wood block is relatively large, which indicates that fungi leave a smaller resource in case of the availability of larger resources (Fukasawa et al. 2020). Other findings include a study by the authors that shows that a fungus switches to explorative growth (stronger outgrowth from the wood block) with smaller wood blocks that are decomposed more rapidly (Fukasawa and Kaga 2020). And then there are findings that are again approved by this study, that show directional outgrowth from an inoculum towards the direction of the bait, which are interpreted as “directional memory”.
I personally read all these studies with great interest, including this one. And I think they show some interesting ecological patterns. But I do not agree with the interpretation of these findings as “memory” or “decision-making”, and definitely not with the term fungal “behaviour” as stated here in the title. All these findings can be explained by simple rules of fungal mycelial growth in response to internal resource supply.
Focusing on this study, I would interpret these results not as a “decision” of the mycelium to migrate from the inoculum to new baits in case of late additions of the new bait, but rather of a depletion of the original wood inoculum after longer time periods. I hypothesize that a control without a bait (unfortunately missing in this study), in which the inoculum would have been taken to new substrate after 56 or 140 days would have shown the same result. After a longer time period, the fungus dies off following complete depletion of the available resources. So the findings do not necessarily point towards a “behavioural decision” as stated by the authors, but may simply relate to the duration of the experiment.
The discussion about the general validity of the term “fungal behaviour” can be left aside when deciding about acceptance of this manuscript. Since there are many studies about this topic published in high ranking journals by high ranking authors, I would suggest to leave this as a general discussion for the future. But I would still argue that the authors need to defend the validity of their statement on fungal “behavioural decision” based on this study, which in my opinion would need an appropriate control with fungi growing without baits controlling for time, to make this argument.
Accept from this major issue, I only have few smaller comments listed below. I also included these points again in this list to clarify.
Minor comments:
L84: I guess DH2O is short for dest. water. It is not a common abbreviation, so please write dest. or add in brackets the meaning. It may be confused with deuterium
L90: Please write g L-1 as unit.
L93: the blocks were on the petri dishes during the whole growth period? That is not fully clear from the text. Please specify
L93: concerning the level of replication, was each wood block placed on a separate petri dish? Or did 20 wood blocks fit into one or two petri dishes?
L102: Maybe more out of curiosity, the experiment does not need to be sterile? Is there no problem by contaminations from other Basidiomycetes in that soil?
L117: So if I understand correctly, after addition of bait (bait and wait) independent of timing fungi were grown another 42d before transferring to new plates. That means in case of "bait" you have a total growth period of 56 days, and in case of "wait" of 136 days. I see a problem here to control for the timing.
As the authors conclude, in case of waiting with the bait, the fungus rather "decides" to leave the inoculum wood block, or migrate. One could also argue after a longer time period, all resources the fungus could us in this wood block were depleted, and thus it either dies off at this part of the mycelium or transfers/recycles this part of the mycelium to a new resource source
I understand this is difficult to control for. But one could have done an additional experiment without baits for example, with different time spans, and checked for outgrowth from the inoculum after different periods. In that case it would not be "migration" or a "decision to leave", but simply depletion of all available resources
L178: I would be interested whether hyphal coverage corresponds to hyphal density and also biomass. It is known that fungi show explorative growth in nutrient scarce environment, which means the mycelium may be larger, but thinner (less biomass) due to the explorative outgrowth of few hyphae.
I think it would add to the discussion to add a paragraph on this method of scanning fungal coverage, how well it relates to biomass values. Either in the discussion part, or in the methods part
Fig. 8: I think this graph clearly shows the relationship among resource depletion and a lack of regrowth. So it is not necessarily related to migration or the timing of the bait
L273: to clarify my point. It is difficult to state that this pattern was migration. It rather indicates that fungi did not survive in the inoculum wood block this long, which would also have happened without a new bait. So the clear connection stated between the timing of adding a bait, and the "decision" to migrate is not proven by these data. It rather relates to the age of a wood block and when the fungus dies off. Only here, there was a new wood block in which it could survive. Is that now behaviour and decision making, or rather a strategy to survive in patchy environments, that is logical based on the explorative growth of a
L390: short concluding section is missing
Author Response
General comments
The manuscript presents novel data on three cord-forming Basidiomycetes, that are grown from wood inocula and exposed to “baits” (=fresh wood blocks) after differing time periods, 14 or 98 days. The authors argue that after a longer period the fungi “decide” to migrate to the new bait and leave the inoculum (which is shown by regrowth from the wood blocks), whereas after shorter periods regrowth occurs from both, inoculum and bait wood blocks.
The manuscript is very well written, and the results analyzed and presented nicely. This study follows several recent studies, partly by the authors, on the topic of “fungal behaviour”, that analysed the growth of cord-forming fungi in relation to inocula and baits, and the “decision making” mostly based on differential sizes of baits and inocula. This study clearly differentiates in its topic based on the novel idea to introduce timing as a factor, and using three fungal species instead of only one.
These previous studies indeed proved some interesting patterns, especially that fungi cannot be isolated from an inoculum wood in case the new wood block is relatively large, which indicates that fungi leave a smaller resource in case of the availability of larger resources (Fukasawa et al. 2020). Other findings include a study by the authors that shows that a fungus switches to explorative growth (stronger outgrowth from the wood block) with smaller wood blocks that are decomposed more rapidly (Fukasawa and Kaga 2020). And then there are findings that are again approved by this study, that show directional outgrowth from an inoculum towards the direction of the bait, which are interpreted as “directional memory”.
I personally read all these studies with great interest, including this one. And I think they show some interesting ecological patterns. But I do not agree with the interpretation of these findings as “memory” or “decision-making”, and definitely not with the term fungal “behaviour” as stated here in the title. All these findings can be explained by simple rules of fungal mycelial growth in response to internal resource supply.
Thank you for your positive comments. We appreciate that there may be semantic conflicts in the concept of fungal behaviour among scientists as it is a novel and developing study field. But we believe that recognizing such a growth response of fungal mycelia to internal and external resource supply is a first step in the study of “mycelial” behaviour as shown in studies on Physarum slime moulds (Dussutour et al. 2010 PNAS). We added these statements in newly added summary paragraph. Please see my response at the end of this response list.
Focusing on this study, I would interpret these results not as a “decision” of the mycelium to migrate from the inoculum to new baits in case of late additions of the new bait, but rather of a depletion of the original wood inoculum after longer time periods. I hypothesize that a control without a bait (unfortunately missing in this study), in which the inoculum would have been taken to new substrate after 56 or 140 days would have shown the same result. After a longer time period, the fungus dies off following complete depletion of the available resources. So the findings do not necessarily point towards a “behavioural decision” as stated by the authors, but may simply relate to the duration of the experiment.
Thank you for valuable comments. We agree that we need wait experiments without bait addition (control) to clarify the effect of bait on mycelial death/alive within inoculum wood, in other words, mycelial decision to migrate. We added a call for further studies including a control wait experiment without bait addition. My hypothesis for this future experiment is that presence of bait accelerates partial death of mycelium within inoculum wood block, i.e., migration. If bait had no effect on partial death of mycelium within inoculum wood block, that is inconsistent with previous paper showing that the size of bait obviously affects the activity of mycelium within inoculum wood block (Fukasawa et al. 2020. ISME Journal).
The discussion about the general validity of the term “fungal behaviour” can be left aside when deciding about acceptance of this manuscript. Since there are many studies about this topic published in high ranking journals by high ranking authors, I would suggest to leave this as a general discussion for the future. But I would still argue that the authors need to defend the validity of their statement on fungal “behavioural decision” based on this study, which in my opinion would need an appropriate control with fungi growing without baits controlling for time, to make this argument.
Leaving aside the general validity of the term “fungal behaviour” in this case, we agree that the difference in time duration between early-baited and wait experiments is problematic to conclude that our results showed decision of fungal mycelia. We calmed down discussion about decision throughout the text and added a call for further studies including a control wait experiment without bait addition, which you suggested, and we agree that this is necessary to clarify decision of fungal mycelia.
Accept from this major issue, I only have few smaller comments listed below. I also included these points again in this list to clarify.
Minor comments:
L84: I guess DH2O is short for dest. water. It is not a common abbreviation, so please write dest. or add in brackets the meaning. It may be confused with deuterium
We changed “DH2O” to “Distilled H2O”.
L90: Please write g L-1 as unit.
Corrected.
L93: the blocks were on the petri dishes during the whole growth period? That is not fully clear from the text. Please specify
Yes. We made it clearer in the text.
L93: concerning the level of replication, was each wood block placed on a separate petri dish? Or did 20 wood blocks fit into one or two petri dishes?
We placed each inoculum wood block on a separate dish in soil microcosm experiment, but in the colonization period, we placed 10 blocks in a dish of agar plate.
L102: Maybe more out of curiosity, the experiment does not need to be sterile? Is there no problem by contaminations from other Basidiomycetes in that soil?
We did not sterilize the soil in the microcosm because the presence of soil microorganisms as a competitor for the focal strains is important for mycelial migration from inoculum to bait. If soil was sterilized and only a focal strain was living in the microcosm, mycelium do not need to leave original inoculum even after it lost its nutritional value to defend. We observed Chaetomium globosum and Trichoderma spp. colonizing on some of the wood blocks. We added a related discussion in the third paragraph in 4.1
L117: So if I understand correctly, after addition of bait (bait and wait) independent of timing fungi were grown another 42d before transferring to new plates. That means in case of "bait" you have a total growth period of 56 days, and in case of "wait" of 136 days. I see a problem here to control for the timing.
As the authors conclude, in case of waiting with the bait, the fungus rather "decides" to leave the inoculum wood block, or migrate. One could also argue after a longer time period, all resources the fungus could us in this wood block were depleted, and thus it either dies off at this part of the mycelium or transfers/recycles this part of the mycelium to a new resource source
I understand this is difficult to control for. But one could have done an additional experiment without baits for example, with different time spans, and checked for outgrowth from the inoculum after different periods. In that case it would not be "migration" or a "decision to leave", but simply depletion of all available resources
We agree that the difference in time duration between early-baited and wait experiments is problematic to conclude that our results showed decision of fungal mycelia. However, since a fungal mycelium has a modular body design, partial growth and partial death is an essential mechanism of mycelial migration in soil microcosm as well as in forest floor (Boddy 2009 Mycoscience). We already discussed about the effect of time duration on the results in the third paragraph of Discussion. We calmed down discussion about decision in this paragraph, and added a call for further studies including a control wait experiment without bait addition, which you suggested and we agree that this is necessary to clarify decision of fungal mycelia.
L178: I would be interested whether hyphal coverage corresponds to hyphal density and also biomass. It is known that fungi show explorative growth in nutrient scarce environment, which means the mycelium may be larger, but thinner (less biomass) due to the explorative outgrowth of few hyphae.
I think it would add to the discussion to add a paragraph on this method of scanning fungal coverage, how well it relates to biomass values. Either in the discussion part, or in the methods part
Hyphal coverage is not a measure of colony area, and representing hyphal density and biomass by colour brightness on photo images. We added following statement in 2.4 Image analysis section:
“Hyphal coverage had been used as a measure of hyphal biomass on soil as it represents hyphal density in unit area (Boddy 1993, 1999, 2009).”
Fig. 8: I think this graph clearly shows the relationship among resource depletion and a lack of regrowth. So it is not necessarily related to migration or the timing of the bait
We agree that we need wait experiments without bait addition to clarify the effect of bait on mycelial death/alive within inoculum wood. We added a call for further studies including a control wait experiment without bait addition.
L273: to clarify my point. It is difficult to state that this pattern was migration. It rather indicates that fungi did not survive in the inoculum wood block this long, which would also have happened without a new bait. So the clear connection stated between the timing of adding a bait, and the "decision" to migrate is not proven by these data. It rather relates to the age of a wood block and when the fungus dies off. Only here, there was a new wood block in which it could survive. Is that now behaviour and decision making, or rather a strategy to survive in patchy environments, that is logical based on the explorative growth of a
Since a fungal mycelium has a modular body design, partial growth and partial death is an essential mechanism of mycelial migration in soil microcosm as well as in forest floor (Boddy 2009 Mycoscience). Leaving aside the general validity of the term “fungal behaviour” in this case, we agree that the difference in time duration between early-baited and wait experiments is problematic to conclude that our results showed decision of fungal mycelia. We already discussed about the effect of time duration on the results in the third paragraph of Discussion. We calmed down discussion about decision in this paragraph and throughout the text, and added a call for further studies including a control wait experiment without bait addition, which you suggested and we agree that this is necessary to clarify decision of fungal mycelia.
L390: short concluding section is missing
We added a short concluding section at the end of the Discussion:
“To summarise, our results showed a possibility that timing of resource addition affects mycelial migration behaviour. We appreciate that there may be semantic conflicts in the concept of fungal behaviour among scientists as it is a novel and developing study field. But we believe that recognizing such a growth response of fungal mycelia to internal and external resource supply as a behaviour is a first step in the study of mycelial behaviour. The results raised a new question that which of available resource amount and time duration affect mycelial behaviour, which should be tested in the future. Concerning the direction of the regrowth of the fungi, a trend of growth from the inoculum side linked to the bait was noticed for one fungal strain only, which does not allow any strong general conclusions to be drawn on memory of fungal mycelia.”

Round 2
Reviewer 2 Report
Thank you for thoroughly addressing the points raised in the review.
Just a small comment on the changes made. In L343/344 in the discussion the authors write "In addition, to evaluate the effect of bait on mycelial migration more accurately in wait experiment, control dishes without bait are also needed." The position and phrasing of this sentence is not optimal, since it puts into question the complete discussion of the results. I would suggest to discuss this point more carefully, also explaining why a control would be needed, and how theses results can already be interpreted in current form. So the reader can interpret this statement better.
Author Response
Just a small comment on the changes made. In L343/344 in the discussion the authors write "In addition, to evaluate the effect of bait on mycelial migration more accurately in wait experiment, control dishes without bait are also needed." The position and phrasing of this sentence is not optimal, since it puts into question the complete discussion of the results. I would suggest to discuss this point more carefully, also explaining why a control would be needed, and how theses results can already be interpreted in current form. So the reader can interpret this statement better.
Thank you for valuable comments. We moved this sentence to the summary section with some additional discussions about how interpret the result in current form.
